# Exploring the associations between narcissism, intentions towards infidelity, and relationship satisfaction: Attachment styles as a moderator

**Ahmet Altınok**[1]*, **Nurseven Kılıç**[2]

**1** Department of Psychology, Experimental Psychology, University of Groningen, Groningen, The Netherlands, **2** Department of Psychological Counselling and Guidance, Eskişehir Osmangazi University, Eskişehir, Turkey

* a.altinok@rug.nl

**Data Availability Statement:** All relevant data are within the manuscript and its Supporting Information files.

## Abstract

The ultimate goal of this research was twofold: (1) to investigate the associations between narcissism, intentions towards infidelity, and relationship satisfaction; and (2) to explore the moderating effect of attachment styles on the link between intentions towards infidelity and narcissism. The findings revealed that the link between narcissism and relationship satisfaction is fully mediated by intentions towards infidelity. Similarly, the full mediating effect of relationship satisfaction exists in the association between narcissism and intentions towards infidelity. Mediational analyses further revealed that narcissism is a predictor of intentions towards infidelity, and this link is moderated by preoccupied, fearful, and dismissive attachment styles. As the results indicate, narcissism plays a significant role in young adults' intimate relationships, and attachment styles have a moderating role in narcissism's effect on romantic relationships. Results and implications are discussed in light of the relevant research findings.

## Introduction

There is a growing consensus among researchers that narcissism increases throughout the world parallel with time [1, 2]. These studies, despite reflecting a widespread concern for narcissism in general, point to a gap in the related literature that indicates a lack of focus on the specific profile of narcissists in intimate relationships. The term "narcissism" is defined as normal and pathological narcissism [3–5]. Normal narcissism means that we are all narcissistic to some extent, and this is natural for individuals. Escalated normal narcissism is generally identified as rating higher scores than the mean on the Narcissistic Personality Inventory (NPI) [6]. However, pathological narcissism refers conventionally to a specific personality disorder: Narcissistic Personality Disorder, or NPD [7]. In the present study, we refer to normal narcissism, which is often measured with the NPI or other similar measures. Individuals with high scores in narcissism are incapable of connecting with others, generally treating people as objects just

**Funding:** The authors received no specific funding for this work.

**Competing interests:** The authors have declared that no competing interests exist.

to facilitate their own wants and needs [8]. In the reports of narcissists' romantic attractions, it can be seen that they are specifically attracted to persons who are (a) of a high social status (e.g., successful, famous, and/or attractive) and can offer the narcissist self-enhancement, and (b) admiration can increase the self-perceptions of narcissists directly through flattery and attention [9]. Moreover, narcissists are typically less interested in warm, close, communal, and caring relationships, and they perceive relationships as arenas for bolstering themselves without regard to their partners [10].

Narcissists' romantic relationships are reported to be transitory, which means lacking in commitment [11]. Likewise, according to Campbell and Foster [12], the link between narcissism and commitment to the romantic relationship partner indicates a negative correlation. That is, narcissism is associated with a game-playing love style, low commitment, and infidelity [2]. Prior research has also shown that narcissists in long-term romantic relationships demonstrate low levels of commitment, are susceptible to infidelity, and have a greater number of divorces than nonnarcissists [13, 14]. If empirical researchers want to draw attention to the low-commitment attributions of narcissists, addressing the causes and consequences of this issue may help them to comprehend the manifestations of narcissists' reduced commitment in ongoing romantic relationships. In this sense, shifting the focus into the investment model of commitment [15, 16], which comprises elements such as satisfaction, investment, and perceived alternatives, may clarify the mechanisms playing a role in infidelity. Satisfaction here refers to the rewards in relationships with respect to the costs, where increased satisfaction leads to greater commitment. Investment means the individuals' efforts to create stakes in the relationship. To exemplify, investments may include shared bank accounts or dwellings, children, shared friendship networks, and even moments. The last element of the investment model, perceived alternatives, refers to the options outside of the relationship. In sum, greater satisfaction and investment lead to greater commitment whereas greater perceived alternatives lead to lesser commitment [15, 16], and lower commitment may contribute to infidelity [17].

Current research in the area of adult intimate relationships is mostly based on attachment theory [18, 19]. The relationship patterns formed with the caregiver-infant relationship during earlier periods are internalized, and they establish and yield the bases on how the individual may initiate relationships with others as well as how he/she may retain such relationships [20]. These mental representations, called internal working models, are rather resistant to change because they operate outside of the conscious awareness realm of the mind. Narcissism can be considered an outcome of the failure to establish a secure attachment between a child and a parent [21]. In other words, narcissists' inability to maintain fulfilling relationships with others is a result of deficient early child-caregiver interactions [22]. Beginning from the early periods, distressed mother-child relationships bring forth the attachment problems associated with them. Some researchers present findings revealing the relationships in the attachment styles-narcissism link (e.g., [21, 23, 24]).

Current related literature shows the relationship between narcissistic personality and relationship satisfaction. Additionally, intentions toward infidelity might play a mediator role in this relationship. Moreover, based on interchangeable prediction, infidelity could have a mediation effect on the relationship between narcissistic personality and infidelity, as we have suggested. Brewer et al. [13] has mentioned that narcissism predicted both previous infidelity experience and intentions to engage in infidelity. Also, Jones and Weiser [25] have found that psychopathy in men and women, and narcissism in women, predicted infidelity in the current relationship. The relationship between narcissism and infidelity could be explained by a lack of commitment and empathy [11], also approval seeking and vulnerability in the relationship [26].

Moving to a more serious relationship status calls for personal and emotional investment, correspondingly it brings more risks, especially for those who are vulnerable to rejection. Thus, narcissistic individuals might have more tendency to cheat due to their vulnerability and approval seeking traits. Initially, it might seem that being popular with more partners could lead to higher relationship satisfaction. However, relationship satisfaction is also related to commitment, intimacy, passion and partner's satisfaction [27].

Furthermore, having several partners might not automatically provide more relationship satisfaction, due to satisfaction relying on much more complicated factors such as: emotional agreement [28], perceived and given support [29], compassion [30], jealousy [31], and spending time together [32]. Ultimately, as narcissism makes staying committed to a relationship difficult, tendencies of infidelity might work to further decrease relationship satisfaction.

In a similar way, narcissism and intentions towards infidelity may be mediated by relationship satisfaction, as low relationship satisfaction specifically increases emotional infidelity [33]. As we expected, intentions toward infidelity play a mediator role in the relationship between narcissism and satisfaction, likewise narcissism and intentions toward infidelity can be mediated by relationship satisfaction.

The present study was carried out in order to reveal the links between narcissism, intentions towards infidelity, relationship satisfaction, and attachment orientations. The first reason why university students were used in the study is that there are empirical findings of the increased narcissism among university students. Twenge et al. [34] report that university students from 31 campuses spread across the US scored progressively higher in narcissism between the early 1980s and 2006. They find a significant and positive correlation between the NPI scores and the year of data collection. Second, as a requirement of the young adulthood period's psychosocial developmental stages, individuals must establish close relationships and experience their maintenance. Fulfilling this stage successfully may influence the quality of the relationships that the individual will establish during the future stages of his/her life. The young adulthood period corresponds to intimacy versus isolation. In this developmental period, the function of the development is to establish "healthy" relations with the social environment [35, 36]. Hence, the fact that this study focuses on early adults is of critical importance. Consequently, in the present study, the aim was to first determine whether the link between narcissism and relationship satisfaction is mediated by intentions towards infidelity, and secondly, to determine whether the link between narcissism and intentions towards infidelity is mediated by relationship satisfaction. Afterwards, the moderating role of attachment styles in the relationship between narcissism and intentions towards infidelity was examined.

## Method

### Participants

Prior to the study, all of the ethical procedures of the study were completed, and the study was approved by Ankara Yildirim Beyazıt University Ethical Committee (449/22.02.2017). The participants contributed to the study with their informed consent. The purposeful sampling method was implemented as one of the dedicated sampling methods. The study group consisted of 407 university students studying in various state universities in Turkey in the spring term of the 2016–2017 academic year. Initially, participants were given detailed information about the research, and they were assured that their identity information would be kept confidential. 177 (43.5%) men and 230 (56.5%) women aged 18–30 years old (SD = 3.58, M = 21.52) participated in the study. 184 of the participants reported being in an ongoing intimate relationship. These students represent a sub-group of the 407 students (the rest were not in a current relationship). The first two hypothesis models (Figs 1 and 2) were tested within this sub-group. In order to be

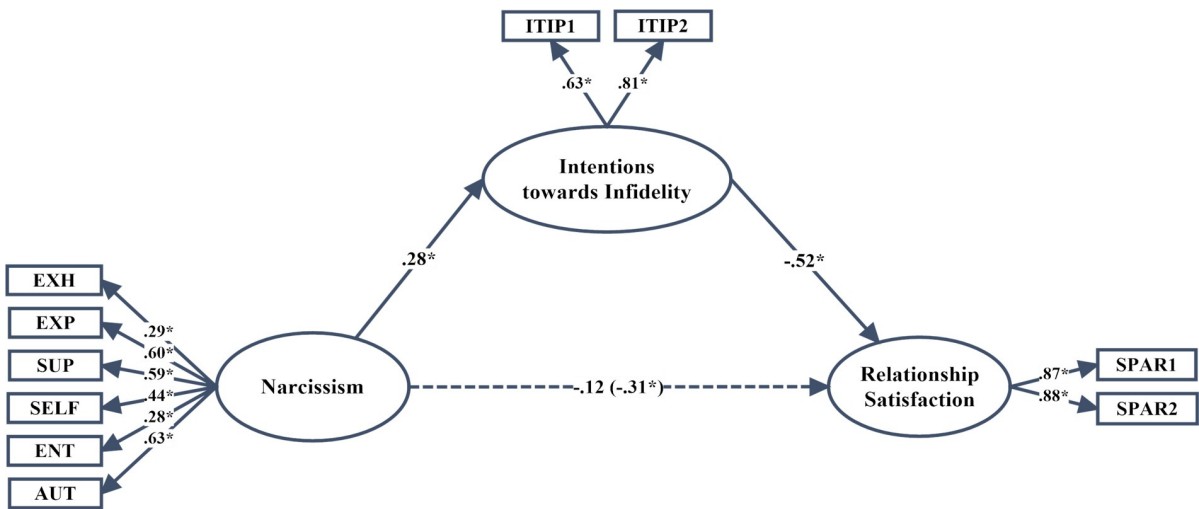

**Fig 1.** *p < .05, standardized parameter estimates of the final structural model; the parameter value given in parentheses was calculated when the effect of the other means on relationship satisfaction was released; EXH: Exhibitionism; EXP: Exploitation; SUP: Superiority; SELF: Self-sufficiency; ENT: Entitlement; AUT: Authority; ITIP1–2 = two parcels from the intentions towards Infidelity Scale; SPAR1–2 = two parcels from the Relationship Satisfaction Scale.

able to use the Relationship Assessment Scale (RAS), we needed obtain the data from individuals in an ongoing relationship and perform individual statistical analyses. Moderating model tests, however, contained the entire sampling of 407 students regardless of whether they were in an ongoing intimate relationship. Therefore, the rest were not asked to complete the RAS, and the third hypothesized structural model was analyzed with the entire sampling.

## Materials and procedure

Data for the hypothetical model tests were collected from the participants through a set of Likert-type questionnaire booklets including the Narcissistic Personality Inventory (NPI), the

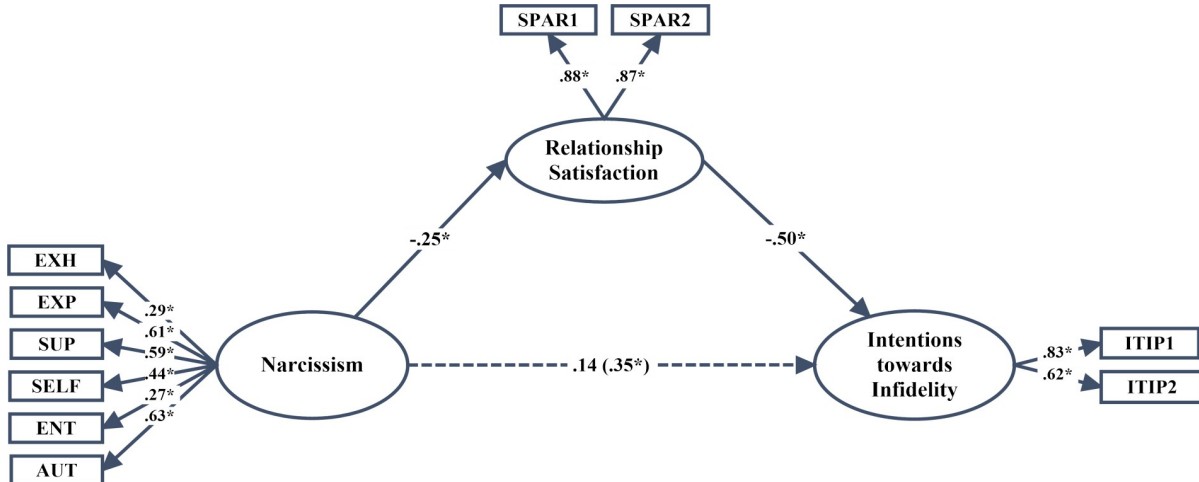

**Fig 2.** *p < .05, standardized parameter estimates of the final structural model; the parameter value is given in parentheses for when the effect of relationship satisfaction on intentions towards infidelity was fixed to zero; EXH: Exhibitionism; EXP: Exploitation; SUP: Superiority; SELF: Self-sufficiency; ENT: Entitlement; AUT: Authority; ITIP1–2 = two parcels from the Intentions towards Infidelity Scale; SPAR1–2 = two parcels from the Relationship Satisfaction Scale.

Intentions towards Infidelity Scale (ITIS), the Relationship Scales Questionnaire (RSS), and the RAS. For the moderating model tests, on the other side, participants completed the same questionnaire booklet except for the RAS. In the model tests, structural equation modelling and maximum likelihood estimation methods were used. Furthermore, in order to analyze the indirect effects on the models, the bootstrapping method was used.

**NPI.** The NPI was developed by Ames, Rose, and Anderson [37] as a 16-item scale that comprises items taken from the larger NPI-40 [6] for the purpose of determining narcissistic personality features. The NPI is the most widely used self-report measure of narcissism. The Turkish adaptation of the NPI was carried out by Atay [38]. The NPI includes 16 items loading on six relational dimensions: exhibitionism, superiority, authority, entitlement, exploitation, and self-sufficiency. Examples of these items include "I know that I am good because everybody keeps telling me so" and "When people compliment me, I sometimes get embarrassed." The participants are presented with 16 statement pairs, and they choose the statement that most accurately applies to their own feelings. The scores of the NPI are rated from 0 to 2 for authority, 0 to 3 for exhibitionism, 0 to 3 for exploitation, 0 to 2 for entitlement, 0 to 2 for self-sufficiency, and 0 to 3 for superiority. Narcissistic responses are coded as 1 and non-narcissistic responses are coded as 0. The total score of narcissism is obtained from a scale of 0 to 16. An increase in the number of scores represents an increase in the level of narcissism [38]. Cronbach's alpha was between .69 and .78 in the original form and .62 in the Turkish version. The internal consistency of the entire inventory in our sampling was .61.

**IT IS.** The ITIS [39] is a one-dimension scale composed of 7 items answered on a 7-point scale of "not at all likely" to "extremely likely." The ITIS measures intentions to be unfaithful; examples of these items include questions such as "How likely are you to be unfaithful to a partner if you knew you wouldn't get caught?" An increase in score indicates a greater intention to engage in infidelity. The Turkish adaptation of the scale was carried out by Toplu-Demirtaş and Tezer [40], and the internal consistency of the scale's Turkish form was reported to be between .70 and .81. The scale demonstrated acceptable reliability in the present study ($\alpha$ = .73).

**RAS.** The RAS was developed by Hendrick [41] to measure individuals' overall satisfaction with their relationship; the Turkish adaptation of the scale was carried out by Curun [42]. Example items include "In general, how satisfied are you with your relationship?" and "To what extent has your relationship met your original expectations?" on a 7-point scale, where 1 = strongly disagree to 7 = strongly agree. Higher scores obtained from the scale indicate greater relationship satisfaction. The internal consistency of the entire scale was .91 in the original form and .86 in Turkish version. The internal consistency of the scale in the current study was .82.

**RSQ.** The RSQ was developed by Griffin and Bartholomew [43] to determine the four attachment styles of secure, fearful, preoccupied, and dismissive. The RSQ consists of 18 items and each item is scored on a 7-point Likert scale between "1: It does not identify me at all" and "7: It identifies me very well." The Turkish adaptation of the RSQ was made by Sümer and Güngör [44] in a cross-cultural comparison, and they stated that the RSQ is reliable to apply to both Turkish and American university students. The internal consistency coefficients of the original form of the RSQ ranged from .41 to .71 and were between .35 and .65 in the Turkish form. The internal consistency coefficients were between .33 and .55 in the present study. Although the RSQ had low internal consistency, the test re-test validity [45] was strong. Furthermore, Griffin and Bartholomew [46] argue that the low internal consistency coefficient was not due to the fact that the subscales were composed of too few items or that the scale had insufficient psychometric quality, but that the subscales included both models of self and others. Additionally, the scale has a good construct and criterion validity.

# Results

## Descriptive statistics

In the current study, analyses regarding the first goal of the study were carried out in two stages. Initially, the correlations between narcissism and intentions towards infidelity were tested (Figs 1 and 2), and afterwards, the moderating effect analysis was carried out. Prior to the model tests, preliminary analyses were carried out to reveal the zero-order correlations, means, and standard deviations among the 10 observed variables, which are reported in Table 1. The skewness values for the observed variables ranged from .11 to 1.08, and the kurtosis variables were between .08 and 1.38. These values demonstrate that the statistical appropriateness of the variables is ensured in terms of normal distribution assumptions.

## Test of the measurement model

In measurement model, the observed variables were designated for the latent variables in the structural model. As the same latent variables (narcissism, intentions towards infidelity, and relationship satisfaction) were included in both models, a single measurement model test was performed. As with the observed variables, the total scores of the NPI subscales served as the latent variable for narcissism. For the intentions towards infidelity and relationship satisfaction latent variables, parcels were assigned proportionally to the number of items since the scales that measure the variables were couched in a one-dimensional structure. This parcelling method was performed by assigning scales to the parcels according to the determined number of parcels, depending on the item-total correlation values.

According to the tests of the measurement model utilized as the model for the study, the goodness of fit values were optimal: $\chi^2(32, N = 184) = 62.91$, p = .001; GFI = .94; AGFI = .89; CFI = .92; IFI = .92; RMSEA = .073 (90% confidence interval for RMSEA = .046–.099). It was determined that the entire factor loads related to the observed variables of all latent variables were high and statistically significant (standardized values ranged from .44 to .91, p < .05; see Table 2).

**Table 1. Correlations, means, and standard deviations among 10 observed variables.**

| Variable | 1 | 2 | 3 | 4 | 5 | 6 | 7 | 8 | 9 | 10 |
|---|---|---|---|---|---|---|---|---|---|---|
| Exhibitionism | - | | | | | | | | | |
| Exploitation | .13 | - | | | | | | | | |
| Superiority | .22** | .33** | - | | | | | | | |
| Self-sufficiency | .01 | .27** | .30** | - | | | | | | |
| Entitlement | .12 | .16* | .15* | .11 | - | | | | | |
| Authority | .16* | .43** | .35** | .29** | .14 | - | | | | |
| Infidelity-P1 | .12 | .02 | .13 | .11 | .31** | .03 | - | | | |
| Infidelity-P2 | .37** | .03 | .16* | .04 | .26** | .05 | .51** | - | | |
| Satisfaction-P1 | -.14 | -.11 | -.20** | -.13 | -.20** | -.06 | -.27** | -.36** | - | |
| Satisfaction-P2 | -.15* | -.07 | -.15* | -.02 | -.19** | -.09 | -.27** | -.37** | .76** | - |
| M | .36 | .52 | .43 | .36 | .29 | .59 | 7.29 | 1.59 | 2.24 | 14.73 |
| SD | .21 | .31 | .37 | .31 | .36 | .38 | 4.51 | 5.11 | 5.79 | 4.69 |

*Note.*

*p < .05

**p < .01, N = 184 (individuals in an ongoing relationship); the Narcissistic Personality Scale subscales: Exhibitionism; Exploitation; Superiority; Self-sufficiency; Entitlement; Authority; Infidelity-P1–P2 = two parcels from the Intentions towards Infidelity Scale; Satisfaction P1–P2 = two parcels from the Relationship Satisfaction Scale

**Table 2. Factor loadings, standard errors, and t-values for the measurement model.**

| Measure and variable | Unstandardized factor loading | Standard error | t | Standardized factor loading |
|---|---|---|---|---|
| Narcissism | | | | |
| Authority | 1.00 | - | - | .62 |
| Entitlement | .44 | .15 | 3.03* | .29 |
| Self-sufficiency | .59 | .14 | 4.37* | .44 |
| Superiority | .95 | .18 | 5.26* | .59 |
| Exploitation | .78 | .15 | 5.28* | .60 |
| Exhibitionism | .26 | .08 | 3.03* | .29 |
| Intentions towards Infidelity | | | | |
| Infidelity-P1 | 1.00 | - | - | .62 |
| Infidelity-P2 | 1.51 | .33 | 4.55* | .83 |
| Relationship Satisfaction | | | | |
| Satisfaction-P1 | 1.00 | - | - | .88 |
| Satisfaction-P2 | .81 | .11 | 7.70* | .87 |

*Note.*

*$p < .01$, Infidelity-P1–2 = two parcels from the Intentions towards Infidelity Scale; Satisfaction-P1–2 = two parcels from the Relationship Satisfaction Scale

In addition, correlations among all latent variables in the hypothesis models were statistically significant (see Table 3).

## Test of the structural models

In this phase of the analysis, two structural models were tested. First, the partial mediation model given in Fig 1 was tested, and the goodness of fit values were found to be close to perfect: $\chi^2$(32, N = 184) = 62.91, p = .001; GFI = .94; AGFI = .89; CFI = .92; IFI = .92; RMSEA = .073 (90% confidence interval for RMSEA = .046–.099). Additionally, it was determined that the standardized means coefficient between narcissism and relationship satisfaction was not statistically significant (β = -.13, p>.05). However, when the effect of intentions towards infidelity was fixed to zero, the standardized means coefficient between narcissism and relationship satisfaction was found to be statistically significant (β = -.31, p < .01). The path between narcissism and relationship satisfaction was subsequently excluded from the model in order to test the fully mediated model, and the goodness of fit values were calculated as $\chi^2$(33, N = 184) = 64.57, p = .001; GFI = .93; AGFI = .89; CFI = .92; IFI = .92; RMSEA = .072 (90% confidence interval for RMSEA = .046–.098). The chi-square difference test results (1.66, 1: p>.05) indicated that the exclusion of this path from the model did not cause a significant deterioration in the model. The final model after excluding the path is shown in Fig 4. Ultimately, as the model demonstrates, intentions towards infidelity has a full mediating effect on the relationship between narcissism and relationship satisfaction.

**Table 3. Correlations among the latent variables for the measurement model.**

| Latent variables | 1 | 2 | 3 |
|---|---|---|---|
| 1. Narcissism | - | | |
| 2. Intentions towards infidelity | .26* | - | |
| 3. Relationship satisfaction | -.25* | -.50* | - |

*Note.*

*$p < .05$

Second, the partial mediation model given in Fig 2 was tested and the goodness of fit values were found to be close to perfect: $\chi^2$ (32, N = 184) = 62.91, p = .001; GFI = .94; AGFI = .89; CFI = .92; IFI = .92; RMSEA = .073 (90% confidence interval for RMSEA = .046–.099). Additionally, the standardized means coefficient between narcissism and intentions towards infidelity was not found to be statistically significant (β = .14, p>.05). However, when the effect of relationship satisfaction on intentions towards infidelity was nullified, the standardized means coefficient between narcissism and intentions towards infidelity was found to be statistically significant (β = .35, p < .01). Later, the path between narcissism and intentions towards infidelity was subsequently excluded from the model in order to test the fully mediated model, and the goodness of fit values were calculated as $\chi^2$ (33, N = 184) = 64.86, p = .001; GFI = .94; AGFI = .89; CFI = .92; IFI = .92; RMSEA = .073 (90% confidence interval for RMSEA = .046–.098) (chi-square difference test: 1.95, 1: p>.05). This result indicates that the exclusion of this path from the model did not cause a significant deterioration in the model. The final model, shown in Fig 5, indicates that relationship satisfaction has a fully mediating effect between narcissism and intentions towards infidelity.

In terms of the explained variance in the models, narcissism explained the 12% variance of intentions towards infidelity and the 10% variance of relationship satisfaction. Relationship satisfaction and narcissism jointly accounted for the 27% variance in intentions towards infidelity.

## The significance of the indirect effects

The bootstrapping method developed by Shrout and Bolger [47] is used to assess the significance levels of the indirect factors in the models. One thousand bootstrapping samples were created during the model application. For the first model (Fig 1), the prediction level of the indirect effect was calculated as -.29 and -.01 within the confidence level of 95% and for the second model (Fig 2), the prediction level of the indirect effect was calculated as .03 and .23 within the confidence level of 95%. The results indicate that the indirect effects were found meaningful in both of the models.

## Test of moderation

At this phase of the study, the moderating effect of attachment styles in the link between narcissism and the intentions towards infidelity was tested. In order to test the mediating and moderating roles of the variables, we followed the guidelines set out by Baron and Kenny [48] and Anderson and Gerbing [49]. To begin with, zero-order correlations, means, and standard deviations for variables are presented in Table 4, followed by the moderating effect tests.

As Table 4 demonstrates, the correlation values did not reveal high correlation for multiple linearity. Additionally, the skewness values for the variables ranged from .03 to 1.35, and the kurtosis values were between .12 and 1.96. These values indicate that the statistical appropriateness of variables is ensured in terms of normal distribution assumptions.

In analysis of the attachment styles' moderating effects, initially gender, narcissism, and attachment styles, followed by the link between narcissism and attachment styles were taken into the hierarchical regression analysis in that order. Baron and Kenny [48] have suggested that when the effects of the predictive variable and the moderating variables were controlled, then the mutual interaction should be statistically significant. Four separate hierarchic regression analyses were implemented for this purpose; the results are shown in Table 5.

As Table 5 indicates, the fearful (β = -.11, p < .05), preoccupied (β = .14, p < .05), and dismissive (β = -.12, p < .05) attachment styles had a moderating effect on the link between narcissism and intentions towards infidelity. However, secure attachment did not have any

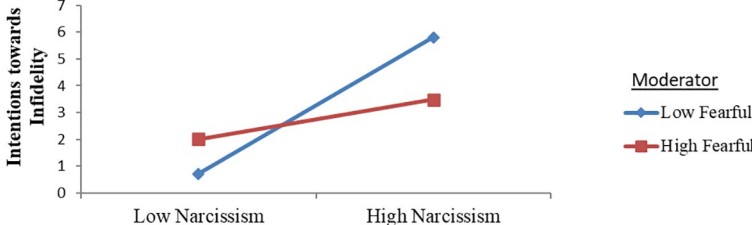

**Fig 3. Intention towards infidelity predicted by the interaction between narcissism and the fearful attachment style.**

moderating effect on this link. When the moderating effect of gender and narcissism were controlled, secure attachment predicted infidelity intentions in a negative way ($\beta$ = -.10, p < .05). This result shows that an increase in secure attachment results in a decrease in intentions towards infidelity.

**Moderating role of fearful attachment style.** Fig 3 illustrating the moderating effect of the fearful attachment style on the interaction between narcissism and intentions towards infidelity demonstrates that individuals with a low level of fearful attachment style reached higher scores for the relationship between narcissism and intentions towards infidelity in comparison to those with a high fearful attachment style. Thus, it can be claimed that an increase in the fearful attachment style causes a decrease in the relationship between narcissism and intentions towards infidelity.

**Moderating role of preoccupied attachment style.** When the moderating effect of the preoccupied attachment style on the link between narcissism and intentions towards infidelity was considered (Fig 4), individuals with a high preoccupied attachment style reached higher scores for the interaction between narcissism and intentions towards infidelity compared to those with low preoccupied attachment levels. This result indicates that an increase in preoccupied attachment level leads to an increase in the association between narcissism and intentions towards infidelity.

**Moderating role of dismissive attachment style.** When the moderating effect of the dismissive attachment style on the interaction of narcissism and intentions towards infidelity was considered (Fig 5), the individuals with a low dismissive attachment style were found to have a higher relationship in their narcissism and intentions towards infidelity when compared to those with high dismissive attachment levels. Therefore, it can be said that as preoccupied attachment decreases, the association between narcissism and intentions towards infidelity increase as well.

## Discussion

In this study, the first prediction was that there is a link between narcissism and relationship satisfaction and that this link is mediated by intentions towards infidelity. Additionally,

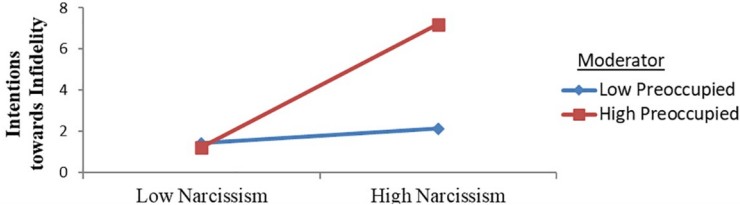

**Fig 4. Intentions towards infidelity predicted by the interaction of narcissism and the preoccupied attachment style.**

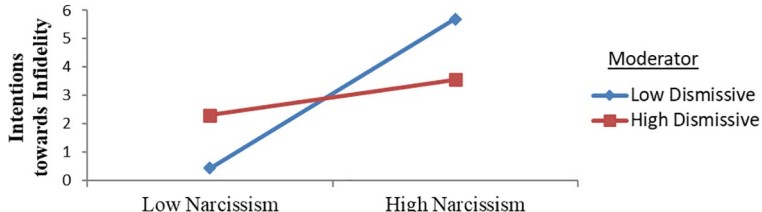

**Fig 5. Intentions towards infidelity predicted by the link between narcissism and the dismissive attachment style.**

relationship satisfaction has a mediator role in the relationship between narcissism and intentions towards infidelity. Further we have suggested that attachments styles have a moderator effects on the relationship between narcissism and intentions towards infidelity.

Consistent with our prediction, the link between narcissism and relationship satisfaction was fully mediated by intentions towards infidelity. That is, narcissism was associated with low relationship satisfaction, and decreased satisfaction was consistently related to infidelity. In other words, this negative association between narcissism and relationship satisfaction was mediated by intentions towards infidelity. As the literature has suggested, narcissistic individuals are less satisfied with long-term relationships, less committed to their romantic partners, and more engaged with infidelity (e.g., [12, 14, 50]).

The second prediction was that there is an association between narcissism and intentions towards infidelity and that this relationship is mediated by relationship satisfaction. Once again, consistent with our prediction, the full mediating effect of relationship satisfaction was revealed in the association between narcissism and intentions towards infidelity. That is, narcissism and intentions towards infidelity were positively correlated, and this correlation was mediated by relationship satisfaction. These results reflect a circle in which narcissists maintain their game-playing love styles. To summarize, relative to nonnarcissists, narcissistic individuals tend to be less committed to their romantic partners and to play games with their romantic partners [51]; they also tend to be less satisfied with their relationships [50] and engage in infidelity more often [52].

Our last hypothesis model was supported by the data, and the findings were consistent with our prediction. We predicted that attachment styles have a moderating role in the association

**Table 4. Correlations, means, and standard deviations among the variables for moderation analysis.**

| Variable | 1 | 2 | 3 | 4 | 5 | 6 | 7 |
|---|---|---|---|---|---|---|---|
| Intentions towards infidelity | - | | | | | | |
| Narcissism | .22** | - | | | | | |
| Gender[a] | -.20** | -.09 | - | | | | |
| Secure | -.04 | .14** | -.15** | - | | | |
| Fearful | -.02 | .05 | .00 | -.32** | - | | |
| Preoccupied | .10* | -.08 | .04 | -.19** | .03 | - | |
| Dismissive | .04 | .15** | -.05 | -.10* | .41** | -.30** | - |
| M | 16.74 | 6.13 | - | 2.57 | 2.50 | 2.48 | 2.78 |
| SD | 7.66 | 2.75 | - | .42 | .53 | .46 | .46 |

*Note.*

$^*p < .05$

$^{**}p < .01$, N = 407

[a] = Women: 1

**Table 5. The hierarchic regression analysis results in which the moderating effects of attachment styles in narcissism and intentions towards infidelity are tested.**

| Moderator | Predictors | Standardized βs | | | |
|---|---|---|---|---|---|
| | | Step 1 | Step 2 | Step 3 | Step 4 |
| Secure | Gender[a] | -.20** | -.18** | -.20** | -.20** |
| | Narcissism | | .21** | .22** | .22** |
| | Secure | | | -.10* | -.10* |
| | Narcissism x Secure | | | | -.02 |
| | $R^2\Delta$ | | .04** | .10* | .00 |
| | $R^2$ | .04** | .08** | .09** | .09** |
| Fearful | Gender[a] | -.20** | -.18** | -.18** | -.18** |
| | Narcissism | | .21** | .21** | .20** |
| | Fearful | | | -.03 | -.03 |
| | Narcissism x Fearful | | | | -.11* |
| | $R^2\Delta$ | | .04** | .00 | .01* |
| | $R^2$ | .04** | .08** | .08** | .10** |
| Preoccupied | Gender[a] | -.20** | -.18** | -.19** | -.19** |
| | Narcissism | | .21** | .22** | .21** |
| | Preoccupied | | | .13* | .15* |
| | Narcissism x Preoccupied | | | | .14* |
| | $R^2\Delta$ | | .04** | .02* | .02* |
| | $R^2$ | .04** | .08** | .10** | .12** |
| Dismissive | Gender[a] | -.20** | -.18** | -.18** | -.18** |
| | Narcissism | | .21** | .20** | .20** |
| | Dismissive | | | .01 | .01 |
| | Narcissism x Dismissive | | | | -.12* |
| | $R^2\Delta$ | | .04** | .00 | .01* |
| | $R^2$ | .04** | .08** | .08** | .10** |

*Note.*

*p < .05

**p < .001, N = 407; β: standardized regression coefficient; $R^2\Delta$: R-squared change

a = Women: 1; $R^2$; the significance values shown above reflect the high significance of the model

between narcissism and intentions towards infidelity. Mediational analyses revealed that narcissism is a predictor of intentions towards infidelity, and this link is moderated by preoccupied, fearful, and dismissive attachment styles. This finding was also aligned with the results of previous studies (e.g., [21, 23, 24]).

It is essential to investigate personality traits and individual differences to gain a more in-depth understanding of why certain people may tend to commit infidelity. It is well known that some aspects of personality are linked to infidelity, such as individuals who are more narcissistic. Thus, as expected, we confirmed both of the proposed models with regard to the first aim of the study with the finding that indicates significant associations between narcissism, intentions to engage in infidelity, and relationship satisfaction. To elucidate, our results indicate that the intentions towards infidelity fully mediated the relationship between narcissism and relationship satisfaction. Moreover, the full mediating role of relationship satisfaction was observed in the relationship between narcissism and infidelity intentions. In this way, we can claim here that our findings are aligned with the results of prior studies revealing the links between narcissism, sexuality, infidelity, low commitment, and poor relationship functioning [13, 17, 25, 53, 54].

In terms of the intimate relationships or relationship commitment of narcissists, we can see that extensive research has focused on the links between narcissistic personality, sexuality, relationship function, and commitment, and studies report significant positive interactions among these traits. In fact, this is what results from narcissists' approach to romantic relationships. In romantic relationships, narcissist individuals look for status and self-esteem instead of intimacy or caring [51], and narcissist individuals turn the concept of love towards the self while nonnarcissists turn this love towards others [12]. Narcissism is associated with poor relationship function such as lack of relationship commitment [12, 50, 55], low emotional intimacy and sexual aggression [56], increased interest in sexual processes [54], and high levels of infidelity engagement [12, 57]. Narcissistic romantic partners are less faithful, less emotionally intimate, less inclined to link sex with intimacy, and eager to have multiple sexual partners [50]. Thus, a specific examination of the role of narcissistic personality traits with various samplings to find out the potential underlying mechanisms in the temptation to be unfaithful to a partner and to predict the attributes of narcissists in intimate relationships may contribute to the literature.

It is worth mentioning that the importance of satisfaction in relationship contexts cannot be underestimated. Most research has emphasized the significance of relationship satisfaction to avoid susceptibility to infidelity, and our results support the idea the literature suggests. In fact, a great number of factors can affect the romantic relationship satisfaction of partners. In this regard, the relationship contexts most strongly associated with susceptibility to infidelity involve sexual dissatisfaction and certain conflicts between partners [52], and some studies have found that low relationship quality is associated with infidelity [50]. Narcissism is also reported to be negatively correlated with measures of relationship quality on the basis of the perspectives of narcissists' romantic partners [58]. Ye et al. [2] have suggested that narcissism has significant negative interactions with both self- and partner-reported relationship satisfaction. Rather than warmth and intimacy ideals, relationships meeting attractiveness and success ideals are more satisfying for narcissists [59]. In addition, studies carried out on satisfaction also report the role of demographic factors, which might be one of the components that modifies the level of satisfaction. For instance, Mark, Janssen, and Milhausen [60] point out that relationship satisfaction is more prominent for women than for men. Similarly, gender is the most commonly studied variable to find out personal differences in the likelihood of committing infidelity. Some studies state that men engage in infidelity more than women (e.g., [61]); however, recent research suggests that women engage in as many acts of infidelity as men [62]. Our findings indicate that women are less likely to be inclined towards infidelity than men.

As the literature has shown, individuals with certain personality attributes such as high narcissism and an insecure attachment style are more likely to commit infidelity (e.g., [17, 52]. To elucidate the underlying mechanisms that clarify why some individuals resist the temptation to be unfaithful while others do not, we hypothesized in the third model that attachment styles might have a moderating role in the relationship between narcissism and intentions towards infidelity. The findings of the model tests highlight the importance of attachment styles to account for the attitudes towards infidelity of individuals with narcissistic personality traits. Our findings reveal the moderating role of preoccupied, fearful, and dismissive attachment styles in the link between intentions towards infidelity and narcissism. In other words, narcissism was a predictor of individuals' temptation to be unfaithful, and attachment styles had a moderating role in this relationship. In this respect, we might suggest that our findings are consisted with literature. For instance, Ahmadi et al. [21] have found a negative relationship between secure attachment and narcissism and positive relationships between avoidant and ambivalent attachment and narcissism. Additionally, Tsang-Feign [63] has shown that attachment avoidance is significantly correlated with vulnerable narcissism, and it is also significantly

and negatively correlated with marital satisfaction. It is not surprising to see that attachment avoidance is related to narcissism because narcissists let themselves be close to others only on a superficial or game-playing level, but then leave the relationship when real commitment is obvious [12]. Additionally, Campbell and Moore [64] have reported that a secure attachment style contributes to relationship satisfaction, which is low in narcissists. Vospernik [65] has further found that narcissistic vulnerability significantly predicts higher levels of attachment anxiety whereas adaptive narcissism significantly predicts lower levels of attachment anxiety. As a result, our results are aligned with the relevant literature. Our findings indicated that some types of attachment styles (preoccupied, fearful, & dismissive) moderate narcissism.

In sum, the results of the analyses carried out for the first goal of the study demonstrated the full mediating role of attitudes towards infidelity in the link between narcissism and relationship satisfaction. Similarly, the mediating role of relationship satisfaction was revealed in the relationship between narcissism and intentions towards infidelity. Furthermore, once the last hypothesis of the research was tested, the moderating role of attachment styles was observed in the interaction between narcissism and intentions towards infidelity. The results of the current study provide a satisfactory contribution to the literature by means of investigating the associations between narcissism, inclination to infidelity, relationship satisfaction, and attachment styles. The findings of this study may extend the literature by elucidating the underlying mechanisms of these traits to some extent. It is widely known that narcissists are more likely to be unfaithful to their partners, and they report low relationship satisfaction in their romantic relationships if the necessary conditions are not provided to meet their expectations. In addition, the romantic partners of narcissist individuals describe their relationships as unsatisfying. Similarly, relationship satisfaction is negatively correlated to individuals with an insecure attachment style. In this study, we sought to determine which types of attachment style (secure, dismissive, fearful, and preoccupied) correlate to narcissism and which are the attributes of narcissist individuals in intimate relationships (in terms of relationship satisfaction and propensity to infidelity). We concluded that narcissism has positive significant correlations with low relationship satisfaction and a high possibility of intentions towards infidelity. Furthermore, dismissive, fearful, and preoccupied attachment styles (insecure) are the moderating factors in the relationship between narcissism and intentions towards infidelity. In this manner, our study offers significant and valuable results for future research.

On the other hand, the current study presents some limitations worth considering. First, like many other studies on narcissism, this study is based on self-report measures only. The number of participants was restricted to young adults in ongoing romantic relationships and the ones who were not in a romantic relationship were excluded from the two proposed models (given in Figs 1 and 2). Participants not in a romantic relationship only took part in the general assessment of the last model (given in Fig 2). For future studies, it remains advisable to recruit a larger sampling to achieve the goals of the study with more detailed statistical analyses. However, we believe the results of this study can illuminate the ways of other researchers and enable mental health practitioners to come up with specific treatment approaches for these types of individuals.

## Supporting information

**S1 Data.**
(SAV)

**S2 Data.**
(SAV)

## Author Contributions

**Conceptualization:** Ahmet Altınok, Nurseven Kılıç.

**Data curation:** Nurseven Kılıç.

**Formal analysis:** Ahmet Altınok.

**Investigation:** Ahmet Altınok.

**Methodology:** Ahmet Altınok.

**Project administration:** Ahmet Altınok.

**Resources:** Ahmet Altınok.

**Software:** Ahmet Altınok.

**Validation:** Ahmet Altınok, Nurseven Kılıç.

**Visualization:** Ahmet Altınok.

**Writing – original draft:** Ahmet Altınok, Nurseven Kılıç.

**Writing – review & editing:** Ahmet Altınok, Nurseven Kılıç.

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
