## [Decision Letter · Decision Letter 0]

11 Sep 2020

PONE-D-20-19314

Exploring the Associations between Narcissism, Intentions towards Infidelity, and Relationship Satisfaction: Attachment Styles as a Moderator

PLOS ONE

Dear Dr. Altinok,

Thank you for submitting your manuscript to PLOS ONE. After careful consideration, we feel that it has merit but does not fully meet PLOS ONE’s publication criteria as it currently stands. Reviewer 2 raised serious issues (probably the most important is that the manuscript doesn't explain the logic for the mediation); please think whether it is possible to address them.

If so, we invite you to submit a revised version of the manuscript that addresses also other points raised during the review process by Oct 26 2020 11:59PM. If you will need more time than this to complete your revisions, please reply to this message or contact the journal office at plosone@plos.org. Please include the following items when submitting your revised manuscript:

We look forward to receiving your revised manuscript.

Kind regards,

Frantisek Sudzina

Academic Editor

PLOS ONE

Journal Requirements:

2. Please change "female” or "male" to "woman” or "man" as appropriate, when used as a noun."

3. Please ensure that you include a title page within your main document. We do appreciate that you have a title page document uploaded as a separate file, however, as per our author guidelines (http://journals.plos.org/plosone/s/submission-guidelines#loc-title-page) we do require this to be part of the manuscript file itself and not uploaded separately.

Could you therefore please include the title page into the beginning of your manuscript file itself, listing all authors and affiliations

"The funders had no role in study design, data collection and analysis, decision to

publish, or preparation of the manuscript."

5. Please upload a copy of Figures 6, 7 and 8, to which you refer in your text. If the figure is no longer to be included as part of the submission please remove all reference to it within the text.

Reviewers' comments:

Reviewer's Responses to Questions

**Comments to the Author**

1. Is the manuscript technically sound, and do the data support the conclusions?

Reviewer #1: No

Reviewer #2: Yes

2. Has the statistical analysis been performed appropriately and rigorously? 

Reviewer #1: No

Reviewer #2: Yes

3. Have the authors made all data underlying the findings in their manuscript fully available?

Reviewer #1: Yes

Reviewer #2: No

4. Is the manuscript presented in an intelligible fashion and written in standard English?

Reviewer #1: Yes

Reviewer #2: Yes

5. Review Comments to the Author

Reviewer #1: PONE-D-20-19314

Exploring the Associations between Narcissism, Intentions toward Infidelity, and Relationship Satisfaction: Attachment Styles as a Moderator

The goal of this work is to examine associations between narcissism, infidelity intentions, relationship satisfaction and test whether the link between narcissism and infidelity intentions is moderated by attachment insecurity.

This work has strengths. The paper is well-written and tests important questions related to the role of individual differences in relationship outcomes. The author tests their hypotheses using in a student sample, which is appropriate, given the topic. At the same time, a handful of critical issues dampen my enthusiasm for this work.

First, while the author does a nice job of reviewing the literature on narcissism, this review includes extraneous information (that narcissism is a diagnosis and individual difference – it isn’t difficult to tell that the authors are not interested in clinical narcissism. It doesn’t necessitate almost an entire page of explanation). Further, the introduction is repetitive in several places and could be streamlined substantially – I’m thinking what’s there could be condensed by at least a third. More importantly, while the author cites appropriate literature to support the mediation hypotheses, they do necessarily not provide rationale or logic to support them – narcissism is related to lower relationship satisfaction and it makes sense that narcissists would tend toward infidelity. However, the logic for tendencies toward infidelity and satisfaction as meditators for each other is missing. In fact, these hypotheses don’t really make sense, given the authors’ own presentation of the literature. For example, with relationship satisfaction as the outcome, is it really narcissists’ tendency to cheat on partners that makes them dissatisfied? The authors state that those higher in narcissism are unable to connect with others and treat people as objects to facilitate their own needs and wants (page 3). By this logic, is it that they’re dissatisfied in relationships because they cheat? While they might be more likely to cheat, it seems that they should be satisfied to the extent that partners fulfill their needs and desires – it isn’t clear how the tendency to cheat could mediate here. Similarly, yes, it seems completely reasonable that narcissists would cheat, but is it because they are unhappy in their relationships? The authors make an argument that the availability of alternative partners should increase the tendency to cheat – how does this translate to satisfaction as a mechanism? Finally, the discussion of attachment in the introduction is difficult to follow and link to operationalization in the data. The authors refer to internal working models in the introduction (which I think is a smart way to pitch the argument), but then they do not actually present an argument with hypotheses. They just describe attachment and say it’s linked to narcissism, but do not provide any logic at all for a moderation hypothesis, much less the four distinct tests of moderation tested in the results.

Second, the authors present this as a 2-study paper when it is decidedly not. A more accurate representation of these data would be as a single study with tests of mediation on a subset of the sample. Either way, the method should include descriptive information about the relationships in this sample – what was the average length, how serious were these relationships (i.e., casual vs very committed), etc. The method should also provide descriptives for the measures – what were the average levels of narcissism, satisfaction, tendencies toward infidelity, and attachment insecurity in this sample (and for the subsamples)? Without this information, it is difficult to really understand the findings. Also, some of these measures have troubling internal reliabilities – several of the measures have low alpha - this complicates interpretation of these findings.

Third, the findings include a layer of analyses that are unexpected and unjustified (i.e., the analyses of factor loadings for subscales of each measure). Also, the author ignore that they are testing mediation models in cross-sectional data, which does not provide convincing evidence of mediation. Additionally, the ways that these variables are operationalized do not make sense, given the hypotheses – specifically, why would satisfaction with one’s current partner *cause* a general tendency to cheat? Further, because the moderation results were not explicitly hypothesized, I worry about whether they would replicate. I also have concerns about how to interpret those interactions – after reading the discussion, it still isn’t clear what they really mean or how they contribute to our understanding of the role of attachment in the link between narcissism and general tendencies toward infidelity. I simply cannot pin down what these findings really mean.

A couple of minor notes: The use of acronyms in tables make them extremely difficult to read (acronyms may save words, but they introduce unnecessary complexity and load for readers. I gave up attempting to interpret Table 1). I would avoid them at all costs. Also, the manuscript is much longer than is necessary, given the theoretical and empirical scopes of this work. This work could be presented much more succinctly.

Reviewer #2: I really appreciate the opportunity to read this article. The literature review both in narcissism and relationship satisfaction was solid and included the relevant literature. Further, the article was very well written. While the internal consistency was low on some of the scales, I think the authors addressed it the best they could. Plus, these are really hard constructs to measure. The SEM is an appropriate way to measure those latent constructs, and the authors did a good job of clearly explaining the findings. I do believe this paper is important and fills a significant gap in the literature.

I was hoping for a little more than summary of the findings in the discussion section. Maybe to expand this section in more detail? For example, in the intro the authors discussed the cohort effect of narcissism. Could more be said in the discussion about what the findings mean for relationships in the future? What kind of prevention or education is needed to combat this? This is just one example of an area to give a small expansion, and I am not tied to it - but perhaps something more than noting that the study supports previous research.

As far as changes, there is a DSM-V which should be DSM-5; in the references there are some issues with APA 7th style (capitalization in some of the references which are in consistent). For example, look at the Karaaziz reference and the King references.

6. PLOS authors have the option to publish the peer review history of their article (what does this mean?). If published, this will include your full peer review and any attached files.

Reviewer #1: No

Reviewer #2: No

---

## [Author Response · Author response to Decision Letter 0]

28 Oct 2020

We would like to thank each of the reviewers for their invaluable and very useful feedback & comments. We have completed the amendments on the manuscript with the comments of the reviewers and submitted the work to the journal once again. Below are the changes made on the manuscript:

The repetitive places in the introduction part have been taken out of the work to present it more succinctly 

The use of Acronyms in Tables has been avoided. We used the full version of the words to make the Tables more easily understandable.

Instead of presenting the manuscript as a two-study paper, we converted it into a single study with tests of mediation on a subgroup of the sample. 

We made changes in the references considering APA 7th style

We add justification for mediation. 

Introduction was shortened. 

Discussion was shortened.

---

## [Editor Report · Decision Letter 1]

30 Oct 2020

Exploring the Associations between Narcissism, Intentions towards Infidelity, and Relationship Satisfaction: Attachment Styles as a Moderator

PONE-D-20-19314R1

Dear Dr. Altinok,

We’re pleased to inform you that your manuscript has been judged scientifically suitable for publication and will be formally accepted for publication once it meets all outstanding technical requirements.

Kind regards,

Frantisek Sudzina

Academic Editor

PLOS ONE

---

## [Editor Report · Acceptance letter]

5 Nov 2020

PONE-D-20-19314R1 

Exploring the Associations between Narcissism, Intentions towards Infidelity, and Relationship Satisfaction: Attachment Styles as a Moderator 

Dear Dr. Altınok:

I'm pleased to inform you that your manuscript has been deemed suitable for publication in PLOS ONE. Congratulations! Your manuscript is now with our production department. 

Kind regards, 

on behalf of

Dr. Frantisek Sudzina 

Academic Editor

PLOS ONE